# Adult-Type Granulosa Cell Tumor of the Testis: A Report of a Case and a Discussion of the Literature

**Georgios Zervopoulos** [1,*], **Nikolaos Mitsimponas** [2], **Filippos Venetsanos** [2] **and Athanasios Papathanasis** [3]

1. Urologist, Private Practice, 11524 Athens, Greece
2. Medical Oncology Department, Hygeia Hospital, 11524 Athens, Greece; nikosmitsimponas@gmail.com (N.M.); philben2@yahoo.com (F.V.)
3. Gastrenterology Clinic, Agioi Anargyroi Cancer Hospital, 11524 Athens, Greece; thanasispap1990@gmail.com
* Correspondence: georgezervopoulos2012@gmail.com

**Abstract:** Testicular granulosa cell tumors (TGCTs) are rare tumors of sex cord-stromal origin. TGCTs are classified into two main categories, the adult type and the juvenile type. The adult type is extremely rare, with only 93 known cases reported in the literature. Herein, we present a report of a case of a 30-year-old male patient who presented with a testicular mass and underwent radical inguinal orchiectomy; the pathology examination revealed an adult-type granulosa tumor. Additionally, we review the literature to summarize the scientific knowledge of an entity barely described worldwide.

**Keywords:** testis; granulosa cell tumor; adult-type; sex cord-stromal tumor of testis; testicular cancer

## 1. Introduction

Sex cord-stromal tumors (SCSTs) comprise approximately 5% of all testicular tumors, while the rest are of germ cell origin [1]. Sex cord-stromal tumors, which arise from the supporting tissues of the testis, include Leydig, Sertoli, and granulosa cell tumors, as well as the fibroma–thecoma family of tumors and mixed tumors [1,2]. Testicular granulosa cell tumors (TGCTs) are rare tumors of sex cord-stromal tissue. They originate from the epithelial elements of the sex cords. The mechanism leading to the development of TGCTs remains poorly understood due to the rarity of these cases. They consist of granulosa cells and stromal components. They are generally low-grade malignancies, mainly characterized by indolent growth and a low risk of metastasis [1]. GCTs are more frequently discovered in the ovaries, where the adult type is more prevalent. On the other hand, the juvenile type, despite being uncommon, predominates in the testis [2]. Men with juvenile TGCTs never had documented metastatic disease [2]. On the other hand, men with adult GCTs can have metastatic disease (10% to 13% of all documented cases) [2,3]. These tumors tend to be slow-growing neoplasm, although having a possible malignant potential to metastasize to distant sites years after initial diagnosis. There have been cases described that, during follow-up, were diagnosed with distant-site metastases even 10 years after the first radical surgical treatment, though appearing with no progression in this whole period. As a result, men who have undergone orchiectomy require extended follow-up because of the delayed metastatic possibility of these tumors [4]. Due to the rarity of TGCTs, there are several unanswered questions regarding the optimal management of patients with localized or metastatic TGCTs. The aim of the present case report is to document a case of an adult-type GCT and to provide an updated summary of the international literature concerning the reported cases of this type of tumor, an infrequent oddity and a tumor that both urologists and oncologists should have in mind.

## 2. Case Presentation

A male patient, 30 years old, was admitted to our outpatient private office from an internal medicine department with swelling in the left testis that was first noticed two

years earlier and had been growing slowly. The patient did not show any disturbance or anxiety due to the slow growth of the testis and he only decided to visit a doctor when the testis changed in texture upon self-examination. By the time of the examination, the left testis was enlarged, and hard during palpation. The patient's medical history revealed no trauma or signs of inflammation or any other known disease. We performed blood tests and radiological control. Blood tests revealed no indication of inflammation (urine culture was negative and C-reactive protein was within normal values). A-fp (A-fetoprotein), β-hg (β-choric gonadotropin), and LDH (lactate dehydrogenase) values were found to be within the normal range. Initially, we performed a scrotum ultrasound, which revealed a hypoechoic solid mass of 3.8 cm diameter, suspicious of malignancy. A scrotum magnetic resonance imaging confirmed the suspicion of malignancy, showing a mass with septum which was enhancing the intravenous paramagnetic contrast agent. The rest of the imaging control with a CT scan of the chest, abdomen, and brain revealed no sites of metastatic disease or any other pathology.

According to the European Association Guidelines 2022 edition update, radical inguinal orchiectomy was proposed to the patient, and he agreed.

We performed a radical inguinal orchiectomy and the left testis with its components was removed lege artis and sent for pathological examination. The patient had a normal postoperative course with no complications according to the noted Clavien–Dindo score. The pathology report revealed a tumor with septa consisting of large, elongated cells with large, grooved nuclei, suggesting an adult-type testis granulosa tumor with a diameter of 3.8 cm, inhibin-positive. Moreover, the presence of Call–Exner bodies helped to confirm the diagnosis of the adult-type granulosa cell tumor of the testis. There was a proliferation rate of mitosis lower than 3/10 hpf. The other components of the testis, i.e., the rete testis, the tunica albuginea, the epididymis, and the vans deferens, were free of the disease. Serum inhibin type B measured in the patient right after the operation and one month later showed a decrease from 32 pg/dL to 4 pg/dL. Nevertheless, inhibin type A did not show the same downward trend. One month after the surgery, inhibin type A was 10 pg/dL, and 6 months after the surgery, it was 16 pg/dL. At this time point, we conducted a new screening with a CT scan of the chest, abdomen, and brain, which did not reveal any metastatic sites. Nine months after the surgery, we performed a new laboratory control and it revealed inhibin type A at 90 pg/dL. We conducted an FDG-PET/CT. It revealed no site of recurrence nor any distant metastases or lymph nodes. The next follow-up is planned to take place in 3 months with new serum markers and any imaging tests needed.

## 3. Discussion

As far as the histology examination is concerned, about 90% of all testicular neoplasms consist of germ cell tumors (GCTs), while the rest involve sex cord gonadal stromal tumors, malignant lymphomas, secondary tumors, and other very rare new growths.

Testicular granulosa cell tumors are rare neoplasms. They are sex cord-stromal tumors of the testicles (SCSTs). SCSTs can differentiate towards Leydig cells, Sertoli cells, and/or other types of sex cord-stromal cells such as granulosa cells, as in our case [5]. Compared to germ cell tumors (GCTs), SCSTs are substantially less frequent, making up less than 5% of all testicular neoplasms in adults. Leydig cell tumors and Sertoli cell tumors are the most common subtypes among them, accounting for 1–2% and 0.5%, respectively, of all testis tumors. However, prepubertal men have SCSTs at a somewhat higher rate [6].

Granulosa cell tumors resemble ovarian granulosa cell tumors morphologically and are classified into adult and juvenile types, just like granulosa cell cancers in the ovary. The first documentation of an AGCT is credited to Laskowski in 1952.

The adult form is more prevalent in the ovaries, where GCTs are more frequently detected. On the other hand, in the testis, the adult form is less common, and the juvenile type, despite being rare, predominates [2].

Some studies suggest that the formation of granulosa cell tumors is associated with sex chromosome abnormalities and irregular gonadal development. It has been shown

that infants with mixed gonadal dysgenesis or intersexual disorder develop juvenile-type GCTs [7]. Regarding the adult type, the expression of the *FOXL2* gene is supposed to play a leading role in its growth. *FOXL2*, a granulosa cell-expressed gene, regulates granulosa cell fate and ovarian function. A missense mutation of *FOXL2* is vital in the pathogenesis of adult-type ovarian GCTs [8]. Regarding the contribution of this gene to GCT development, studies have shown that this mutation impairs the capability of growth differentiation factor 9, an oncocyte-produced protein, in promoting follistatin transcription in the presence of SMAD3. This can lead to increased cell proliferation due to unopposed activin signaling. In addition, *FOXL2* mutation also can reduce apoptosis; apart from that, it can increase the induction of aromatase (CYP19), which promotes estrogen synthesis [9].

Using the PubMed database and manually scanning the reference lists of earlier studies, only 93 examples of adult-type GCTs in the testis could be found in the literature (Table 1). Not every case report had specific information about every clinical trait. As a result, numerous clinical aspects have to be examined using varied sample sizes.

**Table 1.** Review of the announced adult type granulosa cases and its characteristics [10–57].

| Author | Year | Age (Years) | Tumor Size (cm) | Side of Tumor | Metastatic | Presenting Symptom | Endocrine Symptoms | F/U |
|---|---|---|---|---|---|---|---|---|
| Laskovski | 1952 | 35 | 9 | R | | | gynecomastia | 8.5 |
| Cohen | 1953 | 21 | <1 | L | | incidental | gynecomastia | Ns |
| Massachusetts case | 1955 | 53 | 10 | R | | | gynecomastia | Ns |
| Melicow | 1955 | 52 | 13 | R | | | | Ns |
| Mostofi | 1959 | 41 | 10.1 | L | metastatic | mass, no pain | gynecomastia | DOD 5m |
| Marshall | 1977 | 53 | 10 | NS | | mass, no pain | gynecomastia | 17y |
| Talerman | 1985 | 44 | 3.5 | R | | mass, no pain | | 3y |
| Gaylis | 1989 | 41 | 1.8 | R | | painful mass | gynecomastia | Ns |
| Düe | 1990 | 83 | 18 | L | | | | DOC 1m |
| Nistal | 1992 | 61 | 5 | R | | | | Ns |
| Matoska | 1992 | 26 | 10 | L | malignant | | | 14y NED |
| Sasano | 1992 | Ns | Ns | NS | | | | Ns |
| Monobe | 1992 | 42 | Ns | L | malignant | | | AWD |
| Jiminez-Quintero | 1993 | 57 | 2.5 | R | | | | 3y DOC |
| Jiminez-Quintero | 1993 | 55 | 1.3 | L | | | | 3y NED |
| Jiminez-Quintero | 1993 | 60 | 7 | L | malignant | | | |
| Jiminez-Quintero | 1993 | 39 | 4 | L | | | | NED 1m |
| Jiminez-Quintero | 1993 | 29 | 7.5 | R | malignant | | | Ns |
| Jiminez-Quintero | 1993 | 76 | 0.7 | L | | | | Ns |
| Renshaw | 1997 | Ns | Ns | NS | | | | Ns |
| Renshaw | 1997 | Ns | Ns | NS | | | | |
| Renshaw | 1997 | Ns | Ns | NS | | | | |
| Morgan | 1999 | 49 | 2.6 | L | | | | |
| Al-Bozom | 2000 | 48 | 5 | R | | | | 2y NED |
| Wang | 2002 | 54 | Ns | L | | hydrocele, asymptomatic | | 7m NED |
| Guzzo | 2004 | 33 | 1 | L | | | | Ns |
| Suppiah | 2005 | 51 | Ns | L | malignant | | | Ns |

**Table 1.** *Cont.*

| Author | Year | Age (Years) | Tumor Size (cm) | Side of Tumor | Metastatic | Presenting Symptom | Endocrine Symptoms | F/U |
|--------|------|-------------|-----------------|---------------|------------|--------------------|--------------------|-----|
| Hisano | 2006 | 59 | 15 | L | | | | 6y AWD |
| Arzola | 2006 | 32 | 2.1 | L | | painless mass | | 4y NED |
| Lopez | 2007 | 77 | 4 | L | | | | |
| Ditonno | 2007 | 45 | 6.5 | R | | painless mass | | Ns |
| Gupta | 2008 | 12 | 8 | L | | | | 2y Ned |
| Hammerich | 2008 | 55 | Ns | NS | malignant | | | 1y NED |
| Song | 2011 | 28 | 2.6 | L | | | | 3y NED |
| Hanson | 2011 | 21 | 1 | L | | | | Ns |
| Lima | 2012 | 77 | 2.5 | R | | | | 2y NED |
| Lima | 2012 | 22 | 1 | L | | | | Ns |
| Lima | 2012 | 40 | 2.1 | L | | | | Ns |
| Schubert | 2014 | 78 | 1.3 | L | | painless mass | | |
| Cornejo | 2014 | 49 | 3 | R | | | | 23m NED |
| Cornejo | 2014 | 22 | 3.3 | L | | | | 137m, DOOD |
| Cornejo | 2014 | 50 | 0.9 | L | | | | 48m |
| Cornejo | 2014 | Nd | 3 | L | | | | 59m |
| Cornejo | 2014 | 27 | 1.8 | R | | | | lost |
| Cornejo | 2014 | 87 | Ns | NS | | | | 60m |
| Cornejo | 2014 | 47 | 4.5 | R | | | | 169m |
| Cornejo | 2014 | 51 | 2.1 | R | | | | lost |
| Cornejo | 2014 | 47 | 0.5 | L | | | | 58m |
| Cornejo | 2014 | 55 | 2.1 | L | | | | lost |
| Cornejo | 2014 | 25 | 2 | R | | | | 80m |
| Cornejo | 2014 | 45 | 1.4 | R | | | | lost |
| Cornejo | 2014 | 74 | 3.5 | L | | | | lost |
| Cornejo | 2014 | 27 | 1.4 | R | | | | 104m |
| Cornejo | 2014 | 32 | 4.2 | R | | | | 36m |
| Cornejo | 2014 | 18 | 3.2 | L | | | | lost |
| Cornejo | 2014 | 60 | 6 | R | | | | lost |
| Cornejo | 2014 | 23 | 1.4 | R | | | | lost |
| Cornejo | 2014 | 20 | 4.5 | L | | | | 65m |
| Cornejo | 2014 | 52 | 4.3 | R | malignant | | | 1m |
| Cornejo | 2014 | 14 | 0.8 | R | | | | 24m AWD |
| Cornejo | 2014 | 21 | 3 | R | | | | lost |
| Cornejo | 2014 | 17 | 1.5 | R | | | | 37m |
| Cornejo | 2014 | 64 | 2.4 | R | | | | lost |
| Cornejo | 2014 | 14 | 3.5 | R | | | | 28m |
| Cornejo | 2014 | 28 | 2.6 | L | | | | 35m |
| Cornejo | 2014 | 51 | 3 | L | | | | lost |

**Table 1.** *Cont.*

| Author | Year | Age (Years) | Tumor Size (cm) | Side of Tumor | Metastatic | Presenting Symptom | Endocrine Symptoms | F/U |
|---|---|---|---|---|---|---|---|---|
| Cornejo | 2014 | 63 | 1 | L | | | | lost |
| Cornejo | 2014 | 25 | 5.5 | R | | | | 25m |
| Cornejo | 2014 | 44 | 1.2 | L | | | | 1m |
| Cornejo | 2014 | 50 | 4.7 | R | | | | 1m |
| Köksal | 2003 | 40 | 3.5 | R | | 8-month mass | | 20m |
| Mitra | 2008 | 25 | 0.5 | L | | scrotal pain 2 weeks | | lost |
| Kucukodaci | 2008 | 21 | 1.5 | L | | incidental | | 1 year |
| Harrison | 2009 | 65 | 5 | L | malignant | | | DOD |
| Miliaris | 2013 | 37 | 4.2 | L | | painless mass | | 2 years |
| Norman | 2013 | 68 | Ns | R | | | | 18m |
| Rane | 2014 | 34 | 5 | R | | | | 6m |
| Rane | 2014 | 46 | 65 | L | | | | 2.5 years |
| Rane | 2014 | 46 | 11 | R | malignant | | | 3m |
| Tanner | 2014 | 22 | 0.6 | L | | | | Ns |
| Tsitouridis | 2015 | 29 | 2.5 | R | | | | 1 year |
| Giulianelli | 2015 | 80 | 6 | R | | mass, no pain | gynecomastia | 1 year |
| Vallonthaiel | 2015 | 43 | 6 | L | | painless mass | | 1 year |
| Gomez-Valcarcel | 2016 | 23 | Ns | NS | | | | Ns |
| Mohapatra | 2016 | 57 | 4.7 | L | malignant | incidental | gynecomastia | 32m NED |
| Al-Alao | 2016 | 48 | 1.2 | L | | | | Ns |
| Bani | 2017 | 20 | 2 | R | | | | 4m NED |
| Elbachiri | 2017 | 40 | 5.5 | L | malignant | painless growing mass | | 2 years NED |
| Mezentsev | 2017 | 74 | 5 | R | | painless mass | | Ns |
| Meilan | 2017 | 59 | 3.3 | NS | | | | Ns |
| Nunes-Carneiro | 2017 | 31 | 4.5 | NS | | | | 10 years |
| Dieckmann | 2019 | 22 | 0.8 | L | | scrotal discomfort | | Ns |
| Kabore | 2021 | 64 | 1 | L | | painless testicular swelling | | Ns |
| Present report | 2022 | 30 | 3.8 | L | | painless testicular swelling | | 10m |

The juvenile variety of granulosa cell tumors is the most prevalent (though uncommon) testicular tumor in infancy and is only infrequently encountered in adults. Granulosa cell tumors of the juvenile type do not exhibit the usual Call–Exner body development, tend to have a bit more cytoplasm, and do not have the pronounced grooving of the adult variety. The cells are instead grouped in solid sheets, nests, or nodules and frequently form ectatic areas filled with eosinophilic or basophilic material that resemble enormous follicles [58]. Infants under six months old are the age group in which these tumors are most frequently found [59].

Although malignant behavior has rarely been described, adult-type granulosa cell tumors of the testicles tend to grow slowly and have a good prognosis [2,60].

The adult type of granulosa cell tumor shares the same histological characteristics as ovarian granulosa cell tumors. Known as typical Call–Exner body formation, the cells are organized to form solid sheets and microfollicular structures and feature elongated, grooved nuclei and little cytoplasm [60].

In our patient, the pathology review revealed a neoplasm tumor with septa consisting of large cells with large, elongated, grooved nuclei, suggesting an adult-type testis granulosa tumor with a diameter of 3.8 cm, inhibin-positive. Additionally, the presence of Call–Exner bodies helped to establish the diagnosis of an adult-type granulosa cell tumor of the testis.

The age of the present patient is 30 years, which is clearly lower than the median age of 43.2 years and even lower than the majority of the interquartile range (26–55 years), as observed in the literature survey.

From our review, the median tumor size at diagnosis was 3.2 cm. Our patient at the time of diagnosis presented with a tumor of 3.8 cm in the largest dimension. Regarding the laterality of AGCT, our patient presented with a left-sided tumor, which agrees with the findings in the majority of cases reported in the literature, where 46 (50% of cases) had left-sided tumors as opposed to 37 (40% of cases) with right-sided tumors. For 10 (10% of cases) patients, there is no side location information.

Our analysis of the literature revealed that a painless scrotal lump or enlargement is the most typical presenting symptom in patients with adult TGCTs. Overall, like with testicular germ cell tumors, a palpable mass is likely the typical presenting sign of AGCT [3,61]. Gynecomastia and other endocrine-related symptoms are uncommon in these patients; however, occurrences of the aforementioned symptoms in patients with adult-type GCTs have been documented. More specifically, gynecomastia or other endocrine symptoms were observed in 9 of the 93 reported instances (9% of reported cases). This symptom is not specific to AGCT because it is frequently observed in Leydig cell tumors and sometimes also in beta-human chorionic gonadotropin-secreting germ cell tumors [62].

Metastatic disease was never observed in men with juvenile TGCT but in men with adult TGCT [2,3]. The most common site of metastasis is the retroperitoneal lymph nodes, but other sites including the lungs, liver, bone, and inguinal lymph nodes can also be affected [2]. In our review, 12 of the 93 reported cases (13%) were metastatic. Crogg et al., in a recent analysis of the literature, defined some predictive factors for metastases. Predictive variables for metastatic disease include tumor size, angiolymphatic invasion, and the presence of gynecomastia [2]. Our patient was not metastatic at the initial staging, despite the increased level of inhibin type A. We performed a thorough follow-up with control of inhibin type A, which revealed a remarkable upward trend (from 10 pg/dL to 90 pg/dL within 3 months). We performed an FDG-PET/CT, which revealed no site of distant metastases. At the time of writing, the patient remains with no recurrence 9 months after the initial radical surgical treatment and under strict follow-up.

For patients with stage I illness who have undergone orchiectomy as a curative procedure and who do not have high-risk pathologic characteristics, observation is preferred over additional therapies [63–65].

According to two single-institution retrospective series, patients with clinical stage I illness have a decent prognosis. One study found that 38 men were successfully treated with just primary tumor resection. None had progressed to metastatic illness after a seven-year median follow-up. The results of a second cohort of 48 male patients with short-term follow-up showed that individuals with one or no high-risk characteristics could be safely monitored without retroperitoneal lymph node dissection (RPLND) [64,65].

For patients with clinical stage IIA disease or those with two or more high-risk features on orchiectomy, we suggest early RPLND rather than observation. These risk factors can predict the risk of occult metastatic disease at the time of orchiectomy and include

tumor >5 cm, >3 mitoses per high-powered field, positive margins, rete testis invasion, lymphovascular invasion (LVI), cellular atypia, and necrosis [65].

A five-year occult metastatic disease-free survival rate (DFS) of 48% was seen in patients whose tumors showed two or more risk indicators. A five-year occult metastatic DFS of 98% was seen in patients with malignancies that exhibit fewer than two risk indicators [65]. Five patients with stage I illness and at least two risk factors did not experience recurrences in another observational analysis of seventeen patients with testicular SCSTs treated with retroperitoneal lymph node dissection [65]. These risk projections give an indication of the characteristics of metastatic potential, but they may have limitations because of the retrospective nature, the small size of the reported series, and the lack of a central pathology evaluation.

If RPLND is declined, patients at high risk for metastases based on this risk classification may be given more frequent surveillance imaging. For patients with testicular SCSTs, there are no specific recommendations for such surveillance imaging. One plausible strategy is to adhere to a surveillance schedule comparable to that of patients with nonseminoma germ cell cancers of a comparable stage who have received comparable treatment. When advanced illness is found, RPLND and/or metastasectomy may be suggested.

Our patient had a stage I disease without any risk factor and therefore, we suggested observation with a narrow follow-up with imaging due to the elevated levels of inhibin A.

## 4. Conclusions

In conclusion, in this case presentation, we report a case of an adult-type germ cell tumor of the testis and we provide a summary of all reported cases in the international medical literature. This is a very rare entity in adult men with specific histopathological characteristics, specific clinical characteristics, and, in most cases, benign clinical behavior. Nevertheless, there are some cases with metastatic disease even many years after the initial diagnosis, a fact that underlines the need for proper oncological follow-up and detailed staging even in tumors that predominantly have benign clinical behavior.

**Author Contributions:** Conceptualization, G.Z. and N.M.; methodology, G.Z.; software, N.M.; validation, F.V., G.Z. and N.M.; formal analysis, N.M.; investigation, N.M.; resources, A.P.; data curation, F.V.; writing—original draft preparation, G.Z.; writing—review and editing, G.Z. and N.M.; visualization, A.P.; supervision, G.Z.; project administration, G.Z. All authors have read and agreed to the published version of the manuscript.

**Funding:** This research received no external funding.

**Institutional Review Board Statement:** Not applicable.

**Informed Consent Statement:** Informed consent was obtained from all subjects involved in the study.

**Data Availability Statement:** Not applicable.

**Conflicts of Interest:** The authors declare no conflict of interest.

## Abbreviations

| TGCTs | Testicular granulosa cell tumors |
|-------|----------------------------------|
| AGCT  | Adult granulosa cell tumor       |
| CT    | Computed tomography              |
| LDH   | Lactate dehydrogenase            |

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
