# Peer review of "Adult-Type Granulosa Cell Tumor of the Testis: A Report of a Case and a Discussion of the Literature"

_2673-4397, doi:10.3390/uro3030019_

Round 1

Reviewer 1 Report

The paper is well done and very informative. I have only few methodological comments:

- please remove from the discussion the description of the review research and create a specific paragraph (Methods). Please improve the decsription of the literature research.

- please include some histopathologic pictures

- please include a table summarising the literature research

Author Response

Dear Sir, 

we have revised the discussion, though not made a new "method" paragraf, is this of great importance to you? As far as the table is concerned, there has already been attached a table with all the literature's described cases characteristics, if is not displayed please let me know to submit it again.

Histopathologic pictures are  available, I am very sorry about that.

Thank you

Georgios Zervopoulos

Reviewer 2 Report

I would like to congratulate the authors on their work. 

It is true that the case is interesting and rare. It would be appreciated if the authors could provide some pictures, including those taken during the operation or some MRI images. A few images would make the case more interesting, in my opinion. 

Secondly, I wonder if there is any explanation for the upward trend of Inhibin A after surgery in the absence of a tumor. A rise in Inhibin A was also observed immediately following surgery. It would have been interesting to know what the preoperative value was. If possible, I suggest that the authors provide more information regarding Inhibin A's upward trend.

As a final note, after an overall English checkup, I am confident that the manuscript can be published.

It is possible to improve the English.

Author Response

Dear Sir,

we will attach two mri pictures of the tumor.

The inhibin A levels are still under surveillance, no evidence of tumor recurrence is identified till now.

Thank you

Georgios Zervopoulos